# A Rapid, Simple, Trace, Cost-Effective, and High-Throughput Stable Isotope-Dilution Liquid Chromatography–Tandem Mass Spectrometry Method for Serum Methylmalonic Acid Quantification and Its Clinical Applications

**DOI:** 10.3390/diagnostics12102273

**Published:** 2022-09-20

**Authors:** Lizi Jin, Zhenni Liu, Weiyan Zhou, Jie Zeng, Minhang Wu, Yu Zhang, Tianjiao Zhang, Falin He, Chuanbao Zhang

**Affiliations:** 1National Center for Clinical Laboratories, Institute of Geriatric Medicine, Chinese Academy of Medical Sciences, Beijing Hospital/National Center of Gerontology, Beijing 100730, China; 2Chinese Academy of Medical Sciences and Peking Union Medical College, Beijing 100730, China; 3Zhejiang Biosan Biochemical Technologies Co., Ltd., Hangzhou 310012, China

**Keywords:** liquid chromatography–tandem mass spectrometry, methylmalonic acid, vitamin B12 deficiency, method improvements

## Abstract

**Highlights:**

What are the main findings?

What is the implication of the main finding?

**Abstract:**

Background: Methylmalonic acid (MMA) is an essential indicator of vitamin B12 (VB12) deficiency and inherited metabolic disorders (IMDs). The increasing number of requests for MMA testing call for higher requirements for convenient MMA testing methods. This study aims to develop a convenient quantification method for serum MMA. Methods: The method was established based on the stable isotope-dilution liquid chromatography–tandem mass spectroscopy (ID-LC-MS/MS) technique. The LC-MS/MS parameters and sample preparation were optimized. Specificity, sensitivity, robustness, accuracy, and clinical applicability were validated according to CLSI C62-A guidelines. MMA levels in VB12-sufficient subjects and VB12-deficient subjects were measured. Results: MMA and its intrinsic isomer, i.e., succinic acid (SA), were completely separated. The average slope, intercept, and correlation relationship (R) with 95% confidence intervals, during the two months, were 0.992 (0.926–1.059), −0.004 (−0.012–0.004), and 0.997 (0.995–0.999), respectively. The limit of detection and quantification were <0.058 μmol/L and 0.085 μmol/L, respectively. Intra-run, inter-run, and total imprecisions were 1.42–2.69%, 3.09–5.27%, and 3.22–5.47%, respectively. The mean spiked recoveries at the three levels were 101.51%, 92.40%, and 105.95%, respectively. The IS-corrected matrix effects were small. The VB12-deficient subjects showed higher MMA levels than VB12-sufficient subjects. Conclusions: A convenient LC-MS/MS method for serum MMA measurement was developed and validated, which could be suitable for large-scale MMA testing and evaluating MMA levels in VB12-deficient patients.

## 1. Introduction

Methylmalonic acid (MMA), an abnormal metabolic product of defective cobalamin metabolism and methylmalonate, is considered to be a specific diagnostic biomarker of vitamin B12 (VB12) deficiency [1,2,3,4] and methylmalonic acidemia [5]. VB12 (i.e., cobalamin) is a key cofactor of the enzymatic conversion of methylmalonyl-CoA to succinyl-CoA and the conversion of homocysteine to methionine [6]. An insufficiency of VB12 can lead to elevated methylmalonyl-CoA and homocysteine, causing high levels of MMA and total homocysteine (tHcy) in the blood [3,4]. VB12 deficiency is a common and serious condition [3,4,7] with a high prevalence across many populations, for example, 70–80% in African and Asian children, 40% in Latin American children and adults, and close to 20% in the elderly [3,7]. The measurement of MMA level is also helpful in the screening, confirmation diagnosis, and therapy monitoring of inherited metabolic disorder (IMD) [8]. 

Early identification of VB12 deficiency is vital for early determination of causes and early prevention and remission of serious presentations; VB12, tHcy, and MMA are all helpful diagnostic indicators for these disorders. The VB12 assay is the most commonly used test. However, previous studies have suggested that MMA is a more sensitive and representative biomarker for VB12 deficiency than VB12 and tHcy [3,4], because: (1) Testing for tHcy is less specific since its concentrations can be easily influenced by many environmental factors [3,9]; (2) VB12 is a less stable indicator and is easily affected by environmental factors [10], while MMA is very stable [11]; (3) MMA is more sensitive than VB12 because MMA can increase before VB12 falls [12,13]. (4) Elevated MMA can persist for several days even after replacement is started [14]. (5) VB12 measurements cannot reflect the true status of VB12. VB12 is measured by automated immunoassays based on competitive-binding immune chemiluminescence [3,15,16]. However, immunoassays lack specificity since they simultaneously measure VB12, as well as holohaptocorrin and holotranscobalamin [3,10]. Under these circumstances, increasingly more requests for MMA testing, and therefore, effective and convenient MMA testing is in demand.

MMA is a “mass spectrometry (MS)-based disease biomarker” [17]. Currently, there are no routine economic immunoassays that can quantify MMA. Gas chromatography–mass spectroscopy (GC-MS) [18] was the gold standard for MMA measurements. However, GC-MS-based MMA testing has several drawbacks (including high costs, cumbersome and laborious procedures, large sample volume, and long analytical time) [19,20]. Liquid chromatography–tandem mass spectroscopy (LC-MS/MS) methods are currently preferred for MMA measurements [14,21,22,23,24,25,26,27,28,29,30,31,32,33,34,35,36,37,38]. 

There are many published papers on MMA measurements by LC-MS/MS, using derivatized, non-derivatized, more or less complicated sample extraction, etc. Table 1 summarizes the reported analytical procedures of MMA measurement based on LC-MS/MS (from 2000 to 2022 [14,21,22,23,24,25,26,27,28,29,30,31,32,33,34,35,36,37,38]). Given a comprehensive consideration of all the method procedures as well as the aspects of cost-effectiveness, testing speeding, convenience, method performance, purpose, degree of difficulty for introducing the testing, robustness, accuracy, safety, and environmental protection, it is evident that all of these methods have their advantages and disadvantages. The improvement history of MMA testing shows that the method procedures basically consist of one or more of the following steps: multistep solid-phase extraction (SPE), derivations, protein precipitation, and ultrafiltration, while to complete the above step, at least one of the processes of evaporations, incubations, heating, dryings, reconsititutions, or centrifugations is required [14,21,22,23,24,25,26,27,28,29,30,31,32,33,34,35,36,37,38]. Improvements in procedures have usually focused on reducing these steps by changing different reagents or materials or by applying new materials [14,21,22,23,24,25,26,27,28,29,30,31,32,33,34,35,36,37,38]. These improvements have reduced time and cost. However, with the deepening understanding of the clinical significance of MMA and the widespread popularization of mass spectrometry technology in clinical laboratories, requests for MMA testing are evidently increasing, which calls for more convenient testing methods. 

Different from the previous paths of improvement, we took advantage of a simple mobile phase strategy to improve simple MMA detection, which required fewer reagents and a smaller sample. There was no need for complex and dangerous derivation reagents, costly/not always accessible ultrafiltration materials, or processes that are time-consuming and laborious, such as evaporations, incubations, dryings, and reconstitutions. The established method in this study demonstrated several advantages for MMA detection, for example, convenient, environmentally friendly, economical, and more cost-effective for the assessment of VB12 deficiency; only several simple reagents in small volume were needed and sample preparation could be completed in 20 min. Since LC-MS/MS testing is currently mainly manual, such improvements are significant, especially, when there are numerous requests for MMA testing.

## 2. Materials and Methods

### 2.1. Chemicals

Methylmalonic acid (1.0 mg/mL in acetonitrile, 1 mL ampule, certified reference material, Cerilliant^®^, purity 99%) was purchased from Sigma-Aldrich (Burlington, MA, USA). Isotope-labeled internal standard (IS) methylmalonic acid-^13^C_4_ in acetonitrile solution (certified reference material, CAS: 1173019-21-0, product no. M-173-1ML, purity 99%) was purchased from Sigma-Aldrich (TX, USA). HPLC-grade methanol, acetonitrile, and isopropanol were purchased from Fisher Scientific (Waltham, MA, USA). Formic acid was purchased from Honeywell (San Francisco, CA, USA). Deionized water (18 Ω) was produced from a Milli-Q Advantage system (Millipore Corp., Bedford, MA, USA).

### 2.2. Samples

Pooled and individual serum samples for method establishment and individual specimens for clinical application were obtained from fresh leftover specimens in the Department of Laboratory Medicine, Beijing Hospital, Beijing, China. The collection of leftover sera was approved by the Ethics Committee of the Beijing Hospital. 

### 2.3. Calibrators, Internal Standard (IS), and Quality Control (QC) Materials Preparation

Working standard solutions of MMA (218.69 ng/g) used for calibration and working standard solutions of MMA-^13^C_4_ (148.95 ng/g) were prepared gravimetrically in water. All solutions were aliquoted into 2 mL brown ampoules and stored at −80 °C. For each sample batch, the calibrators were freshly prepared. Eight working calibrators were prepared using 100 μL working standard solutions and subsequently diluted with 800, 700, 600, 500, 400, 300, 200, and 100 μL water, respectively. 

Standard solutions of MMA (2003.60 ng/g) used as QC additive standard solutions were prepared gravimetrically in water. To make low-, medium-, and high-levels of QC materials, 3 μL, 5 μL, and 8 μL of QC additive solutions were gravimetrically added to three gravimetrically prepared 400 μL serum pools, respectively.

### 2.4. Sample Preparation

Fifty microliters of IS working solution, 50 μL of samples/calibrators, and 300 μL methanol were added to a 2.0 mL Eppendorf tube. The mixture was vortexed for 20 s and centrifuged at 148,000 rpm, at 4 °C, for 15 min. The upper phase of the mixture was poured into a new 2.0 mL Eppendorf tube and centrifuged at 148,000 rpm, at 4 °C, for 5 min. Fifty microliters of the upper phase was used for the LC-MS/MS analysis. The injection volume was 1 μL. 

### 2.5. LC-MS/MS Conditions

The LC-MS/MS analysis was performed on a 6500 plus triple quadrupole mass spectrometer (AB Sciex, USA) coupled with an ExionLC™ AD ultra-high-performance liquid chromatography system (Applied Biosystems, CA, USA). The Analyst 1.7.2 software (Applied Biosystems, CA, USA) was used for data processing. 

A Shim-pack GIST-HP C18-AQ column (3 μm, 2.1 × 100 mm, SHIMADZU, Japan) with a guard column (Shim-pack GIST-HP (G) C18, 3 μm, 2.1 × 10 with Cartridge (2pcs) and Holder) was used for separation. An isocratic method (100% A phase) was developed for separation. The mobile phase was water containing 0.1% formic acid and 5% isopropanol. The flow rate of the mobile phase was 0.45 mL/min. The column temperature was 35 °C, and the autosampler temperature was set at 8 °C. Acetonitrile was the wash solution of the LC system. The injection volume was 1 μL. A diverter valve was used from 1.9 min to 2.8 min.

The mass spectrometer was operated in the negative electrospray ionization (ESI) mode with multiple reaction monitoring (MRM). The transitions at *m*/*z* 116.9→72.8 for MMA and *m*/*z* 120.9→75.8 for MMA-^13^C_4_ were monitored for quantification. A source temperature of 450 °C and an ion spray voltage of −4500 V were used. Nitrogen gas was used as the curtain gas (CUR), nebulizer gas (GS1), auxiliary gas (GS2), and collision gas (CAD), and the pressures of the gases were set at 35, 35, 45, and low mode, respectively. The declustering potential (DP), entrance potential (EP), and collision exit potential (CXP) were set at −26 V, −12.6 V, and −11.6 V, respectively. The collision energy (CE) was set at −10.9 eV. 

### 2.6. Method Validation

The method performance was validated according to the Clinical and Laboratory Standards Institute (CLSI) C62-A guideline [39]. 

#### 2.6.1. Limits of Detection (LOD) and Limits of Quantification (LOQ)

Fifty microliters of standard solution of MMA (218.69 ng/g) was diluted with water to generate a series of concentrations. The diluted solutions were treated according to the sample preparation described previously. A signal-to-noise ratio (S/N) ≥ 3 with a coefficient of variation (CV) ≤ 20% for 20 injections was defined as the LOD, while an S/N ≥ 10 and a CV ≤ 20% for 20 injections was defined as the LOQ [39,40]. 

#### 2.6.2. Analytical Precision and Recovery

Pooled serum was filtered using a disposable Corning bottle-top vacuum filter with a 0.22 μm membrane and aliquoted. Four pooled samples with different concentrations of MMA were prepared for the precision and recovery evaluation, that is, sample pools (Level 1), low- (Level 2), medium- (Level 3), and high-levels (Level 4) of QC materials described before. All samples were aliquoted into 2.0 mL Corning vials (1 mL/vial) and stored at −40 °C before analysis. Each sample was measured five times per day for five days. 

#### 2.6.3. Matrix Effect

The matrix addition mixing experiment was designed according to CLSI C62A [39] to assess the sample matrix effect before and after IS correction. Two different matrices were prepared: (X) the mixed standard solution of the neat analyte and IS and (Z) the extracted matrix, i.e., the matrix of serum pools (Level 1, Level 2, and Level 3) which were processed based on the established sample preparation procedure. Then, the defined amount of the mixed standard solution (i.e., X) was added to the same amount of extracted matrix (i.e., Z) to prepare the solution (Y), and the same amount of IS solution was added to the extracted matrix (i.e., Z) to prepare the solution (W). The IS in acetonitrile was previously dried under nitrogen gas and reconstituted with the primary mobile phase. The absolute matrix effects and IS-corrected matrix effects were calculated by the following equations:$$\mathrm{The}\;\mathrm{absolute}\;\mathrm{matrix}\;\mathrm{effect}\;(\%)=\left(1-\frac{A_{Y}}{A_{X}+A_{z}}\right)\times 100\%,$$
where *A* is the peak area of MMA;
$$\mathrm{The}\;\mathrm{IS}-\mathrm{corrected}\;\mathrm{matrix}\;\mathrm{effect}\;(\%)=\left(1-\frac{R_{Y}}{R_{X}+R_{W}}\right)\times 100\%,$$
where *R* is the peak area ratio of MMA to MMA-^13^C_4_.

### 2.7. Method Applications

Fresh individual serum samples from healthy controls (*n* = 18, physical examination subjects with normal biochemical indicators, VB12-sufficient subjects (*n* = 24, VB12 > 240 pg/mL, measured by a routine immunoassay), VB12-deficient subjects (*n* = 13, VB12 < 240 pg/mL), patients diagnosed with anemia (*n* = 25), patients diagnosed with vitamin deficiency (*n* = 13), and patients diagnosed with colon cancer (*n* = 11) were randomly collected and measured in a random order. MMA levels in these populations were investigated using the established LC-MS/MS method. One positive urine sample from methylmalonic academia was measured alongside five urine samples from non-IMDs subjects.

### 2.8. Statistical Analysis

Statistical analysis was completed using Microsoft Excel 2016 (Microsoft Corporation, Redmond, WA, USA), SPSS 25.0 (IBM Inc., Armonk, NY, USA), and GraphPad Prism (version 8.0.0 for Windows, GraphPad Software, San Diego, CA, USA). The intra-assay, interassays, and total imprecision were calculated using one-way analysis of variance (ANOVA). Analytical performance specifications were established based on the within-subject (CV_I_) and between-subject (CV_G_) biological variations of serum MMA, i.e., 7.2% and 21.1%, respectively [41]. An allowable imprecision (CV_A_) = 0.75 ∗ CV_I_ was employed to evaluate the precision criterion (i.e., 5.5%). An allowable bias of 10.61%, derived from allowable bias = 0.375 ∗ (CV_I_ + CV_G_)1/2, was used as the bias criterion. A Mann–Whitney U analysis was used to examine the differences in MMA levels of two subjects.

## 3. Results 

### 3.1. Optimization of the Mobile Phase Strategy

MMA is a small polar compound with high retention properties, making the acquisition of good chromatographic behavior of MMA difficult. Searching for a suitable mobile phase component and an isocratic method cost most of the time in the method development. The first mobile phase strategy was 0.1% formic acid water solution (A phase) and methanol (B phase). An isocratic method of 50% A phase was initially chosen and the optimization determined 95% of A phase. This strategy (95% A phase) was friendly to both SA and MMA, but SA had a higher signal than MMA (see Appendix A). Due to the strong retention properties of MMA, isopropanol was added to the A phase and its percentage was optimized from 50% to 2% (5% was adopted). The addition of isopropanol increased the MMA signal. Then, the percentage of A phase was optimized from 50% to 100% (100% was adopted). 

The drying step before the LC-MS/MS analysis was optimized. The dried residue was initially reconstituted by a 5% methanol water solution containing 0.1% formic acid. Interestingly, MMA is a pH-sensitive compound because we found that when the residue was reconstituted by 5% methanol water solution containing 2% formic acid, there was no peak of MMA. Since MMA was monitored in a negative model, a low pH environment could enhance the ion suppression effects leading to a low signal. Reconstitution solutions were optimized from 5% methanol to 100% methanol without formic acid. Better chromatographic behavior, with a smaller peak width (<0.2 min), symmetrical peak, and higher signal of MMA, was obtained when the dried residue was reconstituted with 100% methanol. Thus, the drying step is no longer necessary. 

Notably, since MMA is a high retention compound, the rinse solutions for the LC system were carefully optimized. Acetonitrile, methanol, and isopropanol were chosen for initial optimization. When methanol was employed for the rinse solution, obvious carryover existed after an injection of serum samples. When isopropanol was employed for rinsing, the carryover disappeared but the MMA peak in the next injection also disappeared. When acetonitrile was employed for rinsing, robust and good chromatographic behavior of MMA was obtained and no obvious carryover existed.

The injection volume was determined to be 1 μL, as the obvious peak tailing problems started to be presented when the injection volume was set to more than 1 μL. 

### 3.2. Optimization of Sample Preparation

Protein precipitation was the main preparation step. The type of precipitation reagent and volume of the reaction system were optimized. Acetonitrile, methanol, and isopropanol were chosen for initial optimization while acetonitrile and isopropanol were excluded because their relative spiked recovery rates were less than 80%. Five reaction systems (RS) were initially explored for a good recovery rate and high signals (the large precipitation reagent volume can reduce the signals because of the dilution effects). A relatively good recovery was obtained in RS-5 (see Figure 1).

### 3.3. Method Validation

#### 3.3.1. Chromatographic Separation

The total run time of the LC-MS/MS analysis was 4.0 min per sample. The intrinsic isomer, i.e., succinic acid (SA), can be completely separated from MMA by chromatography. SA had a relatively higher signal than MMA. MMA can be significantly distinguished from SA. The retention time of MMA and SA were 2.21 min and 1.80 min, respectively. Figure 2 presents representative chromatographs of MMA and MMA-^13^C_4_ in standard solutions (A), serum from a healthy control serum (B), and serum from a patient with VB12 deficiency (C). 

#### 3.3.2. Linearity, LOD, and LOQ

The average slope, intercept, and correlation relationship (R) with their 95% confidence interval (CI) obtained from 12 inconsecutive calibration curves used for analysis during two months were 0.992 (0.926 to 1.059), −0.004 (−0.012 to 0.004), and 0.997 (0.995 to 0.999), respectively. The LOD was estimated as <7.05 ng/g (0.058 μmol/L), and the CV for 20 consecutive injections (S/N > 3) was 5.24%. The LOQ was estimated as <10.41 ng/g (0.085 μmol/L), and the CV for 20 consecutive injections (S/N > 10) was 4.14%. 

#### 3.3.3. Precision and Recovery 

Table 2 summarizes the precision and spiked recovery of the LC-MS/MS method at four levels of MMA on five consecutive days. The interrun CV at Level 1 (native pool), Level 2, Level 3, and Level 4 ranged from 3.09% to 5.27%; intrarun CV ranged from 1.42% to 2.69%; and the total CV ranged from 3.22% to 5.47%. All imprecision performace met the allowable precision criterion. The recoveries of MMA at the three levels were 101.51%, 92.40%, and 105.95%, respectively.

#### 3.3.4. Matrix Effect

The absolute matrix effects for Level 1 serum (native, ~0.3 μmol/L), Level 2 serum (spiked, ~0.5 μmol/L), and Level 3 serum (spiked, ~1.0 μmol/L) were −4.74%, −32.95%, and −84.59%, respectively, indicating that ion suppression effects existed in spiked serum without IS correction. IS-corrected matrix effects for Levels 1, 2 and to 3 serum were −0.19%, 6.84%, and 4.06%, respectively, suggesting IS correction could adequately compensate for the observed matrix effect.

### 3.4. Clinical Application

Figure 3 shows the MMA levels in different subjects. VB12-deficient subjects (VB12 < 240 pg/mL) had significantly higher MMA levels than VB12-sufficient subjects (VB12 > 240 pg/mL) (*p* < 0.05). Additionally, elevated MMA levels were observed in some patients diagnosed with anemia (4/25) and vitamin deficiency (2/13). The MMA levels in these populations were investigated using the established LC-MS/MS method. 

## 4. Discussion

In this study, we established an LC-MS/MS method for MMA quantification that had clear advantages over the previously established LC-MS/MS methods. It is trace, simple, fast, cheap, sensitive, accurate, robust, economic, environmentally friendly, and cost-effective for MMA measurement and is more suitable for laboratories that have large request volumes of MMA testing. These improvements were achieved using protein precipitation combined with a simple mobile phase strategy. Good chromatographic separation of MMA and its isomer was achieved by using an isocratic elution strategy, i.e., 100% A phase (deionzed water containing 0.1% formic acid and 8% isopropanol). Materials and reagents, which are complex or not always accessible, and procedures in previous methods, such as derivatization, multistep SPE, incubation, evaporation, drying, or reconstitution, were not required in this MMA quantification method. We believe this LC-MS/MS method for serum MMA is suitable and convenient for the evaluation of MMA status for VB12 deficiency and can be a reference for laboratories intending to improve their established methods as well as laboratories that plan to introduce MMA testing programs. 

The analytical performances of the previous LC-MS/MS methods have been studied [14,21,22,23,24,25,26,27,28,29,30,31,32,33,34,35,36,37,38]. The LOD and LOQ of previous methods have ranged from 0.03–1.95 μmol/L and 0.03–4.20 μmol/L, respectively. The LOD and LOQ of our simplified method were 0.058 μmol/L and 0.085 μmol/L, respectively. The mean recoveries of previous methods varied from 90% to 111%, whereas the mean recoveries of our method ranged from 92.40% to 105.95%. The intra-assay, inter-assay, and total imprecisions of previous methods were 1.3–8.0%, 3.8–8.5%, and 4.6–10.7%, respectively, while the imprecision indicators of our method were 1.42% to 2.69%, 3.09% to 5.27%, and 3.22% to 5.47%, respectively. The LC-MS/MS run time of previous methods varied from 1.0 min to 10 min per sample and the time for sample preparation for a sample ranged from 4 h to 7.5 h, whereas the throughput of our method was 4.0 min per sample with around 20 min for sample preparation.

MMA can be a specific diagnostic biomarker of VB12 deficiency and some inborn errors of metabolism (IEM) [1,2]. It has been reported that moderately elevated MMA (over 0.4 μmol/L in serum, ~46 ng/g) was an early indicator of acquired vitamin B12 insufficiency, and a massive elevation of MMA (over 40 μmol/L in serum, ~4613 ng/g) could strongly indicate IMDs, e.g., methylmalonic acidemia (an IMD with a relatively high prevalence) [22]. We compared MMA levels in VB12-sufficient patients and VB12-deficient patients and observed higher MMA levels in VB12-deficient patients. Additionally, we observed that some patients diagnosed with colon cancer, anemia, diabetes, and coronary disease had elevated MMA. 

One of the limitations of this study and previous studies was the lack of measurements of reference materials which have ceritificed values and uncertainty for truness. Current MMA measurements lack qualified certified reference measurement procedures and reference materials. The standard materials for calibrators and IS that we employed are certified reference materials with certified purities and uncertainties, thus, obtaining metrological traceability to the SI (System International) unit. [34,35]

## 5. Conclusions

In this study, we established a trace, simple, fast, cheap, sensitive, accurate, robust, economic, environmentally friendly, and cost-effective LC-MS/MS method for serum MMA quantification. This LC-MS/MS method can act as an easy assay and provide fast report results to evaluate the MMA status for vitamin B12 deficiency patients, and is especially applicable for large-scale MMA testing.

## Figures and Tables

**Figure 1 diagnostics-12-02273-f001:**
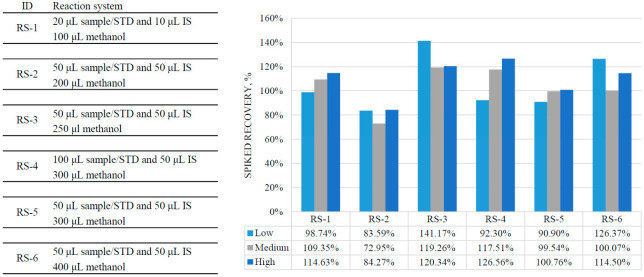
Optimization results of reaction systems in sample preparation.

**Figure 2 diagnostics-12-02273-f002:**
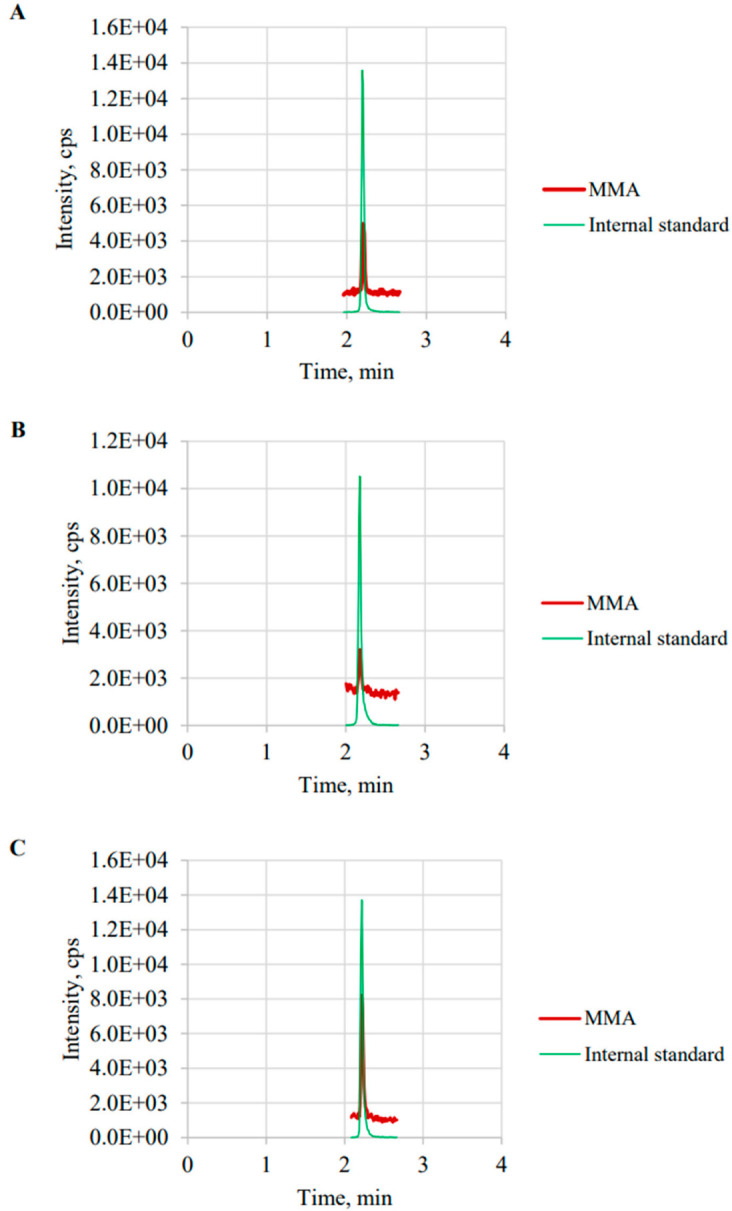
Representative chromatographs of: (**A**) calibrators; (**B**) healthy controls; (**C**) patients.

**Figure 3 diagnostics-12-02273-f003:**
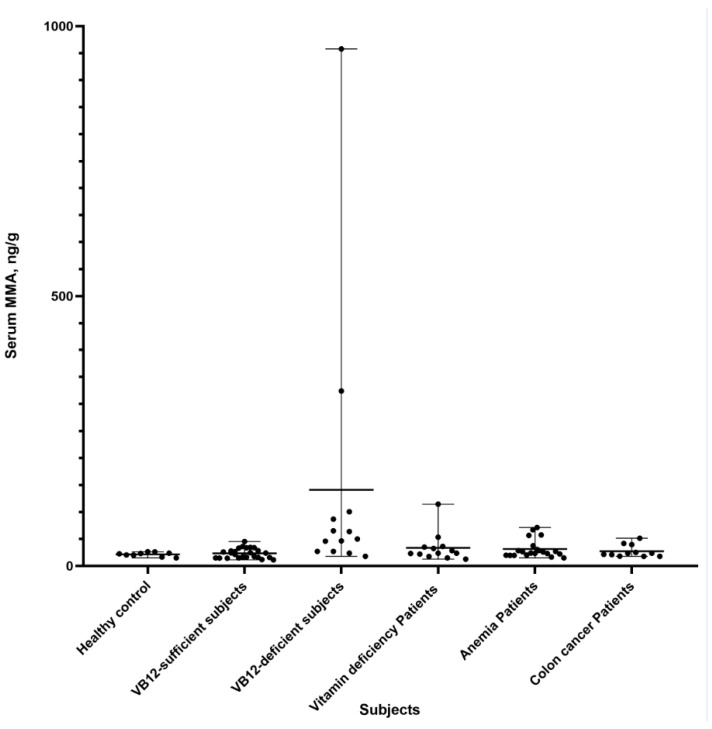
Methylmalonic acid levels in clinical subjects.

**Table 1 diagnostics-12-02273-t001:** A literature review of previous methylmalonic acid quantification based on the liquid chromatography–tandem mass spectrometry method (from 2000 to 2022). Abbreviations of the names of equipment/columns are not listed and abbreviations of reagents are list in the table for reading convenience. Note: LC-MS/MS, liquid chromatography–tandem mass spectrometry; LC, liquid chromatography; MS, mass spectrometry; MMA, methylmalonic acid; LOD, limit of detection; LOQ, limit of quantification; HCl, hydrochloride; SPE, solid phase extraction; SRM, selected reaction monitoring; CVs, coefficient of variations; NA, not available; MRM, multiple reaction monitoring; DBS, dried blood spot; SIM, single-ion monitoring; HPLC, high-performance liquid chromatography; tHcy, total homocysteine.

Study (Year)	Materials and Method Procedures				Precision, Accuracy, LOD, and LOQ	Method Characteristics
	LC-MS/MS Platform	Reagents	Sample Preparation	LC-MS/MS Parameters		
Magera et al. (2000) [21]	API 3000 (Perkin-Elmer Sciex) with two Perkin-Elmer Series 200M pumps	Methanol Water Formic acid 3 mol/L HCl in n-butanol Acetonitrile	Sample volume of 600 μL (100 urine with 500 μL water) with 600 μL of internal standard solution (1.2 nmol/L MMA-d3). Solid phase extraction (SPE) on a Gilson ASPEC automated SPE sample processor, then, preconditioning, sample loading, washing, and elution. Evaporate the elute to dryness in a water bath at 30 °C under dry nitrogen. Transfer the residue using 200 μL methanol and evaporate the methanol in 5 min on a 40 °C dry heating block under dry nitrogen. Derivatization: derivatize the resulting residue using 200 μL 3 mol/L HCl in n-butanol at 65 °C for 15 min and evaporate the excess reagents in 10 min at 40 °C under nitrogen. The residue was dissolved in 100 μL of 80:20 (by volume) acetonitrile/deionized water.	LC: (1) Column: LC-18; (2) Mobile phase: acetonitrile in 1 mL/L formic acid (60:40, by volume). MS: (1) Positive-ion mode; (2) Monitor from *m*/*z* 231.0 to *m*/*z* 119.1 for MMA and from *m*/*z* 234.1 to *m*/*z* 122.0 for MMA-d3; (3) Selected reaction monitoring (SRM) mode.	Inter- and intra-assay CVs were 3.8–8.5% and 1.3-3.3%, respectively, at the mean levels of 0.13, 0.25, 0.60, and 2.02 μmol/L. Mean recoveries were 96.9% (0.25 μmol/L), 96.0% (0.60 μmol/L), and 94.8% (2.02 μmol/L). 0.12 μmol/L. 0.23 μmol/L.	Analytes: MMA only. Validated sample types: serum (but claimed types were plasma and urine) Validated applications: NA. Required procedure steps: SPE, multievaporation, drying, derivatization, multireconstitutions, etc.; complex. Requiring more than two hours for sample treatment (for per sample). Requiring derivatization reagents, etc.
Kushnir et al. (2001) [22]	API 2000 (Applied Biosystems/ MDS SCIEX, Foster City, USA) tandem mass spectrometer with a PE series 200 HPLC system (Perkin-Elmer Analytical Instruments)	Methanol Water Acetonitrile Methyl-tert-butyl ether Phosphoric acid HCl (3 mol/L) in n-butanol Ammonium formate	1 mL for serum/plasma analysis, and 0.1 mL of sample and 0.9 mL of water for urine analysis. Add 100 μL internal standard solution and 3 mL methyl-tert-butyl ether containing 30 mL/L phosphoric acid. Evaporate the solvent, add 40 μL of n-butanol containing 3 mol/L HCl, incubate the mixture at 50℃ for 5 min. Evaporate the excess derivatizing reagent, reconstitute the residues with 75 μL of a mixture of methanol and 0.005 mol/L ammonium formate (1:1 by volume), and transfer them to labeled autosampler vials.	LC: (1) Column: Luna C18; (2) Mobile phase: 850 mL/L methanol and 150 mL/L ammonium formate buffer (0.005 mol/L), PH 6.5. MS (1) Positive-ion mode; (2) Monitor from *m/z* 231 to *m/z* 119 for MMA, and from *m/z* 234 to m/z 122 for MMA-d3; (3) MRM mode.	Within- and between-run CVs were 2.7–4.8% and 4.5–5.8%, respectively, at the levels of 0.3 μmol/L and 1.0 μmol/L. Mean recoveries of 93.9% at 0.3 μmol/L and 99.1% at 1.0 μmol/L. 0.05 μmol/L. 0.1 μmol/L.	Analytes: MMA only. Validated sample types: plasma (but claimed types were serum, plasma, and urine) Validated Applications: NA. Requiring steps: multievaporation, derivatization, heating, reconstitutions, etc.; complex. Requiring 1 mL serum for sample volume. Requiring complex reagents. Requiring at least one hour for sample treatment.
Schmedes et al. (2006) [23]	Micromass Quattro Micro tandem mass spectrometer with a Waters 2795 Alliance HPLC system	d9-Butanol Acetonitrile Methanol Formic acid 37% HCl Water	750 μL serum was mixed with 375 μL internal standard solutions (2 μmol/L). SPE (go through preconditions, sample load, wash, and elution). The column was washed sequentially with 1 mL each of water, methanol, and butanol before elution of the MMA with 300 μL of a 10:90 mixture of concentrated (37%) HCl and butanol. Incubate the elution for 15 min at 70 °C, after 15 min, the temperature was lowered to 54 °C, and then left overnight for evaporation of the HCl-butanol reagent. Evaporate the residues and leave 100 μL of the liquid, add 500 μL acetonitrile-water (20:80 by volume).	LC: (1) Column: Waters Symmetry C18 cartridge; (2) Mobile phase: 1 g/L formic acid-acetonitrile (35:65 by volume) MS: (1) Positive-ion mode; (2) Monitor from *m*/*z* 231 to *m*/*z* 119 for MMA and from *m*/*z* 234 to *m*/*z* 122 for MMA-d3; (3) MRM mode.	Total CVs were 5.0–6.7% at 0.15 μmol/L, 0.36 μmol/L, and 0.65 μmol/L. Mean spiked recovery was 101%. NA. <0.048 μmol/L.	Analytes: MMA only. Validated sample types: plasma Validated Applications: NA. Procedure steps: SPE, multievaporation, derivatization, reconstitutions, etc.; complex. Required sample volume: 750 μL. Requiring at least 1.5 h for sample treatment.
la Marca et al. (2007) [24]	Applied Biosystems/MDS Sciex API 4000™ Triple-Quad Mass Spectrometer equipped with an Agilent 1100 Quaternary Capillary Pump	Acetonitrile Formic acid Water	Punch a 3.2 mm filter paper disk containing ~3.4 μL whole blood from each DBS, and extracted it for 15 min with 200 μL of a solution containing acetonitrile/H_2_O 7:3 and 5 mL/L formic acid, plus 330 nmol/L labeled MMA as internal standard; 2 μL of injection for analysis.	LC: (1) Column: Gemini C6-phenyl(Phenomenex); (2) Mobile phase:an isocratic profile of40:60 between mobile phase of H2O (eluent A) and acetonitrile (eluent B), each containing 5 mL/L formic acid. MS: (1) Negative-ion mode; (2) Monitor from *m/z* 117.1 to *m/z* 73 for MMA and from *m/z* 120.1 to *m/z* 76 for MMA-d3; (3) MRM mode.	Intra- and inter-assay CVs were 3.6–8% and 3.1–6%, respectively. Recovery: 93.6–123.0%. 1.95 μmol/L. 4.20 μmol/L.	Analytes: MMA and 3OH-PA (3-OH-propionic acid). Validated sample types: DBS. Validated Applications: capable of monitoring and quantifying MMA and 3OH-PA during newborn screening as a 2nd-tier test. No validations on vitaminB12 deficiency assessment. Procedure steps: simple. Required sample volume: ~3.4 μL whole blood on DBS. Requiring approximately 18 min for sample treatment. Little known about the ways to/details of method validation. Low sensitivity: the highest LOQ and LOD as compared with the other methods, and is not suitable for the investigation of populations with normal (usually <2 μmol/L) to deficient levels of vitamin B12 (varied MMA levels). Organic solvent in larger volume were required as the mobile phase as compared with our method.
Blom et al. (2007) [14]	Micromass Quattro LC (Waters) with an Agilent HP1100 HPLC (Amsterdam, the Netherlands)	Methanol Formic acid Microcon ultrafilter	An aliquot of 100 μL of plasma was pipetted onto the Microcon ultrafilter, followed by 100 μL 0.8 μmol/L MMA-d3 internal standard solution. After vortexing, the tube was centrifuged for 30 min at 14,000× *g*. Acidified 100 μL ultrafiltrate with 10 μL of 4% formic acid and inject 10 μL of the sample into the LC-MS/MS system.	LC: (1) Column: Symmetry C18; (2) Mobile phase: 15% methanol/0.4% formic acid. MS: (1) Negative-ion mode; (2) Monitor from *m/z* 116.8 to *m/z* 72.9 for MMA and from *m/z* 119.8 to *m/z* 75.9 for MMA-d3.	Intra- and inter-assay CVs at 0.278 μmol/L were 1.5% and 6.7%, respectively. Spiked recovery of 98–101% at levels up to 2 μmol/L. 0.1 μmol/L. NA.	Analytes: MMA only. Validated sample types: plasma Validated Applications: NA. Procedure steps: simple and consist only of ultrafiltration and centrifugations. Required sample volume: 100 μL for plasma. Approximately 35 min for sample treatment. Requiring ultrafilter tubes which are not cheap and not always accessible. The centrifugation requires a longer time for the ultrafiltration. Not suitable for laboratories which have large request volumes. Having the requirement on sample volumes. Not suitable for a sample with a volume less than 50 μL.
Lakso et al. (2008) [25]	Agilent1100 LC/MSD	Acetonitrile Acetic acid Ammonium acetate Water	Human plasma treated with EDTA or citrate (200 μL) was added to 800 μL of the protein precipitation solution in 2 mL glass autosampler vials. The vials were capped, placed on an orbital shaker for 5 min, centrifuged at 6400× *g* for 10 min at 15 °C, and then placed in the autosampler of the LC-MS instrument.	LC: (1) Column: Merck SeQuant ZIC-HILIC; (2) Mobile phase: 4 volumes acetonitrile plus 1 volume 100 mmol/L ammonium acetate buffer, pH 4.5. MS: (1) Negative-ion mode; (2) MMA andD3-MMA were quantified by SIM mode(*m/z* 117.2 and 120.2, respectively).	Intra-assay CVs were less than 5% on all 6 days and inter-assay CVs of single measurements were also less than 5%. Spiked recoveries were between 90% and 93%. 0.03 μmol/L. 0.09 μmol/L.	Analytes: MMA only. Validated sample types: plasma Clinical Applications: no validations on clinical patients. Procedure steps: simple and consists only of protein precipitation and centrifugation. Required sample volume: 200 μL for plasma. Approximately 15 min for sample treatment. Requiring reagents are cheap and easily accessible. The SIM mode is less reliable than MRM mode. In particular, using ammonium acetate buffer would require more time and effort for the equipment maintaince/startup before/after use, and would be required to prevent salting out during use, otherwise, it would easily cause contamination or scrapping of the instrument or column! Organic solvent in large volume were required as the mobile phase (e.g., to prepare 1000 mL mobile phase, 800 mL acetonitrile is required). The total LC-MS/MS assay time, including column washing and reconditioning, was 10 min, which is longer as compared with other methods. LC-MS platform was used, but the LC-MS platform is less reliable than the LC-MS/MS platform. LC-MS/MS is the mainstream platform in current and future clinical laboratories. Not suitable for laboratories that do not have LC-MS platforms.
Carvalho et al. (2008) [26]	Waters QuattroMicro tandemmass spectrometerequipped with an atmosphericpressure chemicalionization(APCI) probe and two ShimadzuLC-10ATvp HPLC pumps	2,3,4,5,6-Pentafluorobenzylbromide (PFBBr) Acetonitrile Dichloromethane Tetrabutylammonium hydrogensulfate (TBAHS) Water	Aliquots of 50 μL of serum were transferred. 10 μL of internal standard solution (MMA-d3) at 100 μmol/L, 50 μL of 0.4 mol/L TBAHS solution (pH 8.8) and 400 μL of 0.1 mol/L PFBBr in dichloromethane were added. The tubes were incubated in an Eppendorf Thermomixer at 85 °C and 1400 rpm for 1 h in a fume hood with exhaustion. The tubes were centrifuged for 10 min at 13,000 rpm. The lower phase was transferred to chromatographic vials, and 50 μL of dimethyl sulfoxide(DMSO) was added to the organic extract. The samples were then subjected to LC–MS/MS analyses.	LC: (1) Column: Synergi-MaxRP, Phenomenex; (2) Mobile phase: (isocratically with 70% acetonitrile in water. MS: (1) Negative atmosphericpressure chemical ionization; (2) Monitor from *m/z* 477 to *m/z* 231 for MMA derivative, and from *m/z* 480 to *m/z* 234 for MMA-d3 derivative; (3) Full scan mode.	Within-day CVs at 0.15 μmol/L, 0.49 μmol/L, 2.15 μmol/L were 7.5%, 6.4%, and 4.2%, respectively. Total CVs at 0.15 μmol/L, 0.49 μmol/L, 2.15 μmol/L were 10.7%, 7.0%, and 4.8%, respectively. Recoveries were between 97% and 105%. 0.03 μmol/L. 0.08 μmol/L.	Analytes: MMA only. Validated sample types: serum. Clinical Applications: NA. Procedure steps: complex, dangerous, etc. Required sample volume: only 50 μL for serum. Requiring more than 1.5 h for sample treatment. Requiring reagents that are complex and dangerous. PFBBr is a lacrimator and an eye irritant.
Fasching et al. (2010) [27]	Waters Acquity LC-MS/MS System (Waters Corp, Milford, MA)	0.9% Saline dialysis solution Water Methanol Formic acid Amicon Centrifree YM30 filter units #4104 (Ionpure Technology, Lowell, MA)	Self-made dialyzed plasma for calibrator preparation (endogenous methylmalonic acid removal, a complex step). Pipette 200 μL of sample into the top portion of filter unit. Tap unit to make sure sample is fully in the unit. Pipette 50 μL of the internal standard solution to the filter unit. Tap unit to make sure volume is fully in unit. Mix well, using a vortex mixer. Centrifuge at 1800× *g* for 40 min at 22 °C. Transfer 100 μL of filtrate to labeled autosampler vials, which contain 10 μL of 4% formic acid. Mix well for analysis.	LC: (1) Column: Waters Symmetry C18 (2) Mobile phase: a mixture from pump A2 (85 %—type 1 water with 0.1% formic acid) and B1 (15 % –methanol with 0.1% formic acid). MS: (1) Negative-ion mode; (2) Monitor from *m/z* 116.9 to *m/z* 72.9 for MMA and from *m/z* 119.9 to *m/z* 75.9 for MMA-d3.	Typical intra- and interassay CVs are <10%. NA. NA. NA.	Analytes: MMA only. Validated sample types: plasma Validated Applications: NA. Procedure steps: simple and not requiring SPE, evaporation, derivatization, dryness, and reconstitutions. Required sample volume: 200 μL for plasma. At least 45 min for sample treatment. Requiring ultrafilters which are not cheap and not always accessible. Using ultrafilter requires long time for centrifugation. Little known about method performance. The self-made dialyzed plasma used for calibration is hard and complex to prepare.
Pedersen et al. (2011) [28]	Initial work was performed on a Micromass Quattro MicroTM atmospheric pressure ionization (API) triple quadrupole tandem mass spectrometer (Waters Corp), and later, transferred to an API 4000 QTrap (AB SCIEX, Foster City, CA) Acquity ultra-high-performance liquid chromatography (UPLC) unit (Waters Corp.,Milford, MA)	1-Cyclohexyluriedo-3-dodecanoic acid (CUDA) Methanol Ammonium formate Succinic acid Phosphoric acid Methyltert-butyl ether (MTBE) 3N HCl in n-butanol Deionized water	Take 25 μL serum spiked with 25 μL of 100 nmol/L MMA-d3. A 400 μL aliquot of 0.5 mol/L H3PO4 in MTBE was added, vortexed for 2 min followed by 3 min centrifugation to separate residual water. A 300 μL aliquot of the supernatant was transferred to tubes and the solvent was evaporated to residue (10–15 min) by centrifugal vacuum evaporation. Residues were reconstituted in 40 μL of 3 nmol/L HCl in n-butanol, vortexed to mix, and incubated in a 60 °C water bath for 30 min. Solvent was removed by centrifugal vacuum evaporation in an acid resistant system, as above. Residues were then reconstituted in 100 μL of 100 nmol/L CUDA in mobile phase and vortexed to assist dissolution. Extracts were transferred to spin filter tubes, spun for 3 min at 4500 rcf, transferred to autosampler vials, capped, and stored at −20 °C until analysis.	LC: (1) Acquity BEH C18 column; (2) Mobile phase: 53% methanol/47% 1.67 mmol/L (pH 6.5) ammonium formate in deionized water. MS: (1) Positive-ion mode; (2) Monitor *m/z* 231 > 119 for MMA; *m/z* 234 > 122 for MMA-d3.	Intra- and inter-assay CVs was under 10%. MMA-d3 surrogate recovery averaged 93 ± 14%. The lowest calibration standards with an effective concentration of 50 nmol/L MMA have an average signal-to-noise of 31. NA.	Analytes: MMA only. Validated sample types: serum. Validated Applications: suitable for the MMA investigation of populations with normal to deficient levels of vitamin B12. Procedure steps: multievaporation, derivatization, reconstitutions, etc.; complex. Required sample volume: only 25 μL for serum. Requiring at least 1.5 h for sample treatment. Requiring reagents that are complex and dangerous, etc.
Yuan et al.(2012) [29]	TSQ Quantum Access triple quadrupole mass spectrometer(ThermoFisher Scientific) with a transcend TLX-4 multichannel HPLC system (ThermoFisher Scientific) A Cyclone-MAX TurboFlow column (50 × 0.5 mm, ThermoFisher Scientific) was used for online extraction, and a mixing column (Agilent, Santa Clara, CA, USA) was placedbetween the injector and the TurboFlow column	Water Methanol Acetonitrile Isopropanol Acetone Formic acid Ammonium acetate Sodium bicarbonate Saline solution	500 μL sample (serum or plasma) was mixed with 500 μL of water and 25 μL of working IS solution. Bond Elute strong anion exchange SPE cartridges were conditioned with methanol (2×), 10 mol/L formic acid (2×), methanol (1×), and water (3×). After sample loading, the cartridges were washed with water (2×) and eluted with 125 μL of 18 mol/L formic acid. Ten microliters of the eluate was injected for analysis.	LC: (1) Column: Allure^®^ Organic Acids column; (2) Turbulent flow and HPLC conditions are complex. MS: (1) Negative ion mode; (2) Two MRM transitions: a quantifier (117.1→73.1) and a qualifier (117.1→55.2), were monitored for MMA. One MRM transition (120.1→76.1) wasmonitored for d3-MMA.	The intra-assay and totalCVs were within 4.6% for all levels tested. 98–111%. NA. 26.2 nmol/L.	Analytes: MMA only. Validated sample types: serum (claimed serum and plasma). Validated Applications: NA. Procedure steps: multistep SPE; complex. Required large sample volume: 500 μL for serum. Requiring approximately 1 h for sample treatment. Requiring to prepare many reagents and solutions for multistep SPE.
Fu et al. (2013) [30]	A triple-quadrupole MS/MS system (Applied Biosystem/MDS SCIEX API 4000 Qtrap) was coupled with a Shimadzu HPLC system and a Leap Technologies auto sampler	Tris(2-carboxyethyl)phosphine hydrochloride (TCEP-HCl) Formic acid Methanol Water Amicon Ultra 0.5 mL 10K Da	100 μL of EDTA or heparinized plasma or serum samples were spiked with 250 μL of HPLC grade water and 100 μL of internal standard solution. Mix the mixture gently for 3 s. 50 μL of TCEP-HCl (1g/34.8 mL in water) was added and mixed gently for 3 s. The mixtures were vortexed twice for 15 s at a medium speed and incubated for 15 min at room temperature. The solutions were transferred with plastic transfer pipette into Amicon Ultra 0.5 mL 10K Da, and then centrifuged at 13,500 rpm for 10 min. The filtrates were transferred into HPLC vials, and 10 μL was injected into the instrument.	LC: (1) Column: reversed-phase C18; (2) Mobile phase: phase A (aqueous 0.2% formic acid) and mobile phase B (0.2% formic acid in methanol). MS: (1)Both positive and negative modes were used. Negative modes for MMA; (2) MRM transition of *m/z* 117.1/73.0 was monitored for MMA and *m/z* 120.1/76.0 was monitored for D3-MMA.	Intra-assay and inter-assay CVs were 2.1–4.9% and 2.7–5.9%, respectively. Recovery: 118.00–120.05%. 0.03 μmol/L. NA.	Analytes: inborn-error biomarkers: MMA, tHcy, methionine, and 2-methylcitric acid. Validated sample types: plasma (claim plasma and serum). Validated applications: methylmalonic acidemia, etc; however, only one vitamin B12 deficiency patient were validated. Procedure steps: simple and consists of ultrafiltration and centrifugations but needs incubation. Required sample volume: 100 μL for plasma. Requiring approximately 20 min for sample treatment. Requiring to prepare additional reagents, i.e., TCEP-HCl. Requiring Microcon ultrafilters which are not cheap and always not accessible. Unsatisfied recovery rate (118–120%). More cost for vitamin-investigating patients since tHcy can be measured by economic immunoassays and other biomarkers are not relevant to vitaminB12 assessment.
Hempen et al. (2015) [31]	Shimadzu high-performance LC (HPLC) system coupled to a Q-Trap 3200 mass spectrometer from Applied Biosystems	Tris(2-carboxyethyl) phosphine hydrochloride (TCEP) Methanol Formic acid Water Microcon centrifugal filter tubes (Millipore)	Mix 100 μL of plasma with 50 μL internal standard and 50 μL TCEP (25 g/L in water) The mixture was vortexed directly, incubated for 30 min and, meanwhile, vortexed again after every 10 min. The following deproteinization of plasma was performed by ultrafiltration using Microcon centrifugal filter tubes. The solution was transferred to the sample reservoir of the Microcon ultrafilter device and the tube was centrifuged at 15 °C for 30 min at 16,300× *g*. Then, the filtrate was pipetted into a vial for LC-MS/MS analysis.	LC: (1) Column: Reversed-phase C18 column; (2) Mobile phase: 5% methanol and 95% of a 0.06 mol/L formic acid. MS: (1) Both positive and negative modes were used, negative modes for MMA; (2) MRM transition of *m/z* 116.9/72.8 was monitored for MMA and *m/z* 119.9/75.9 was monitored for D3-MMA.	The intraday CVs were 3.2% or less and interday CVs ranged from 3.5% to 6.3%. Recovery: 97.8–111.9%. 0.038 μmol/L. NA.	Analytes: MMA and tHcy. Validated sample types: plasma. Validated Applications: no validation. Procedure steps: consists of protein precipitation, and ultrafiltration, a reduction step has to be carried out to ensure the measurement of tHcy, complex. Required sample volume: 100 μL. Requiring at least 1 h for sample treatment. More cost for vitamin deficiency-investigating patients since tHcy can be measured by economic immunoassays. Requiring Microcon ultrafilters which are not cheap and always not accessible.
Kushnir et al. (2016) [32]	Triple quadrupole mass spectrometer AB3200 with TurboVion source (AB Sciex, Foster City, CA) with built-in switchingvalve; binary HPLC pump series 1260 (Agilent Technologies, SantaClara, CA), vacuum degasser, autosampler CTC PAL(Carrboro, NC) equipped with fast wash station	Ammonium formate Formic acid Phosphoric acid Methanol 2-Propanol Methyl-tert-butyl ether (MTBE) K_2_HPO_4_ NaH_2_PO_4_ Sodium chloride 3 mol/L HCL in 1-butanol Water Dialyzed plasma (MMA and SA free).	1 Take 500 μL of serum or plasma, 50 μL of urine sample, and 450 μL of water to the urine; spiked with 50 μL of working internal standard; 2 Set 96-well plate on evaporator and evaporate organic phase(50 °C) until completely dry. 3 40 μL of derivatizing reagent (3 M HCl in 1-butanol) and cover the plate with sealing mat. Incubate the plate at 70 °C for 10 min. 4 Set 96-well plate on evaporator and evaporate organic (50 °C). 5 Add, in each well, 200 μL of reconstitution solvent and cover plate with the sealing mat. Vortex, centrifuge, and inject.	LC: (1) Column: Luna C18; (2) Mobile phase A: 5 mmol/L ammonium formate and Mobile phase B: 5 mmol/L ammonium formate in methanol. MS: (1) Positive ion mode; (2) Monitor from *m/z* 231.2 to *m/z* 119.1 for MMA and from m/z 234.2 to *m/z* 122.1 for MMA-d3.	Total CV of the method wasbelow 10%. NA. NA. NA. 0.1 μmol/L	Analytes: MMA only. Validated sample types: serum (claimed serum and plasma). Validated Applications: NA. Procedure steps: multievaporation, derivatization, reconstitutions, etc.; complex. Required sample volume: 500 μL for serum. Requiring approximately1 h for sample treatment. Requiring multiple reagents which are complex and dangerous. Using ammonium acetate buffer would require more time and effort for the equipment maintaince/startup before/after testing, and would be required to prevent salting out during the testing, otherwise, it would easily cause contaminations or scrapping to the instrument or column.
Ambati et al. (2017) [33]	Agilent 6490 triple-quadrupole mass spectrometer (Agilent, Santa Clara, CA, USA) with an Agilent 1290 seriesbinary pump, online degasser, autosampler, and thermostat column compartment (Agilent Technologies, Waldbronn, Germany)	Formic acid (FA) Water Methanol (CH3OH) Acetonitrile (ACN) ^12^C_6_-3NPH.HCL 1-Ethyl-3-(3’-dimethylaminopropyl) carbodiimide HCl (EDC-HCl) Pyridine compounds 3 kDa Amicon filters and SeQuant ZIC-HILIC column	40 μL of plasma was spiked with 5 μL of internal standard mixture solution. 20 μL of freshly prepared 200 mmol/L ^12^C_6_-3NPH-HCL solution in 50% ACN and 20 μL of freshly prepared 120 mmol/L EDC.HCl in 6% pyridine solution in 50% ACN were added to 40 μL plasma mixture and mixed well. The mixtures were incubated at 40 °C for 30 min and subsequently cooled on ice for 1 min. All the reaction mixtures were diluted with 1.915 mL of 10% aqueous ACN and analyzed by LC-MS.	LC: (1) Column: ACQUITYUPLC CSH C18; (2) Mobile phase: 0.1% FA in H2O (A) and 0.1% FA in ACN (B). MS: (1) Positive ion mode; (2) Monitor from *m/z* 387.2 to *m/z* 178 for MMA and from *m/z* 390.2 to *m/z* 178 for MMA-d3; (3) MRM mode.	Intraday and interday imprecision were 5.2% and 8.9%, respectively. Recovery: 84.3%. 75 nmol/L. 360 nmol/L.	Analytes: inborn-error biomarkers: MMA, malonic acid, and ethylmalonic acid. Validated sample types: plasma. Validated applications: accurate quantitation of MMA, malonic acid, and ethylmalonic acid in plasma of mouse. Procedure steps: derivatization, etc.; complex. Required sample volume: 40 μL for serum. Requiring approximately 35 min for sample treatment. Requiring to prepare many complex/dangerous reagents and solutions, etc.
Ma et al. (2022) [34]	A Waters Acquity I-Class UPLC system (Binary Solvent Manager, Thermostatic Column Manager, and FTN Sample Manager) and a Waters TQ-XS triple quadrupole MS/MS system were used which were controlled by MassLynx 4.2 software (Waters, Milford, MA, USA)	Dithiothreitol (DTT) Methanol Acetonitrile Formic acid Water	A serum sample (100 μL) was mixed with 20 μL of internal standard solution (4.23 nmol/mL). DTT (50 μL, 500 mmol/L) was added and mixed at room temperature for 15 min to completely reduce disulfides. Proteins were precipitated via the addition of 300 μL acetonitrile containing 1% formic acid. After 10 min of centrifugation at 13,000× *g*, 150 μL of the clear supernatant was transferred and concentrated to dryness under nitrogen (requires at least 20 min), and then reconstituted using 100 μL water containing 1% formic acid. Finally, 5 μL was injected into the system for analysis via LC-MS/MS. A urine sample was diluted 10 folds using ultrapure water, and then prepared in the same manner as serum samples without protein precipitation and drying. The sample was finally diluted using 500 μL water containing 1% formic acid, and then transferred to a 700 μL 96-well collection plate.	LC: (1) Column: Premier HSS C18; (2) Mobile phase: Eluent A consisted of ultrapure water and 0.1% formic acid (MS grade) andEluent B consisted of acetonitrile containing 0.1% formic acid (LCgrade). MS: (1) Both positive and negative modes were used; (2) MonitoringMode: not accessible.	Inter-assay CVs of 3.3–7.7% for serum MMA and 2.5–4.6% for urine MMA; total CVs were 7.5–13.6% for serum MMA and 5.9–7.5% for urine MMA. The recovery for serum MMA was 94.6–99.4%, and for urine MMA was 101.6–105.6%. NA. 0.04 nmol/mL.	Analytes: MMA and tHcy. Validated sample types: serum and urine. Validated Applications: suitable for the investigating serum vitaminB12 status. Procedure steps: simple, consists only of protein precipitation and centrifugation, but requires dryness and reconstitutions; time-consuming. Required sample volume: 100 μL for serum. Requiring approximately 50 min for sample treatment. Their clinical validation results indicated that urine tHcy and urine MMA may not be suitable markers for assessing VB12 status. Less cost-effective and more cost for vitamin-investigating patients since tHcy can be measured by economic immunoassays. Due to more time required by dryness, reconstitutions, and limited dryness devices, the method may not convenient or suitable for cases in which large volumes of MMA testing are requested; dryness and reconstitutions for several samples seems rapid but will be cumbersome for numerous samples. The capability of acetonitrile in protein precipitation is weak. The supernatant was not that clear and there were minor flocculent sediment floating when 300 μL acetonitrile was used to precipitated 100 μL serum. The direct injection of supernatant obtained by simple centrifugation may have potential risk of contaminating the mass spectrum or blocking the column. So acetonitrile should not be selected for protein precipitation unless filter tubes is used for centrifugations.
Ueyanagi et al. (2022) [35]	LCMSTM-8040 system (Shimadzu Corp., Kyoto, Japan) coupled with CLAM-2030	Water Formic acid Methanol Acetonitrile 3-Nitrophenylhydrazine (3-NPH) N-(3-dimethylaminopropyl)-N′-ethylcarbodiimide (EDC) HCl	All the sample deproteinization and derivatization reactions were performed using the fully automated LCMS pretreatment system CLAMTM-2030 (Shimadzu Corp., Kyoto, Japan). In CLAM-2030, 20 µL of methanol was dispensed onto a filter. (1) 10 µL of serum sample and 90 µL of methanol containing the internal standard were dispensed into a dedicated vial and stirred for 60 s. (2) Samples were filtered using vacuum pressure for 60 s. (3) The collected deproteinized samples were derivatized by adding 25 µL of 200 mM 3-NPH (50% methanol solution) and 25 µL of 200 mM EDC/9% pyridine (50% methanol solution), stirred for 30 s, followed by incubation at room temperature (RT, 23 °C) for 15 min. The derivatized sample was then transferred to the autosampler of the LC-MS/MS system, and 10 µL were injected.	LC: (1) Column: Shim-pack Scepter HD-C18-80; (2) Mobile phase: (A) 0.1% aqueous formicacid and (B) acetonitrile. MS: (1) Negative ion mode; (2) Monitor from *m/z* 387 to *m/z* 178 for MMA and from *m/z* 391 to *m/z* 151 for MMA-^13^C_4_; (3) Multiple reaction monitoring(MRM) mode.	Intra- and inter-assay imprecisions were 4.8–5.2% and 4.9–9.0%, respectively. Recovery: 97.2–110.4%. 0.063 µmol/L. NA.	Analytes: 19 organic acids, including MMA. Validated sample types: serum. Validated Applications: suitable for the investigating organic acidemias; no validations on vitaminB12 deficiency assessment. Procedure steps: automated but complex steps in the machine, e.g., ultrafiltration, evaporation, derivatization, incubations. Required sample volume: 10 μL for serum. Claimed requires less than 30 min for sample treatment. The adaptability of CLAM-2030 in non-Shimadzu LC-MS/MS system is unclear and may not economic for the evaluation of vitamin B12 deficiency. Requiring multiple reagents which are complex and dangerous.
Boutin et al. (2022, May) [36,37]	Xevo TQ-S micro (Waters Corporation) tande mmass spectrometer, the UPLC system used was an Acquity I-Class (Waters Corporation) equipped with a flow-through needle injector	Water Formic acid Methanol Ammonium hydroxide (28%) NH_4_OH Water-wettable PTFE AcroPrep Advance 96-well filtration plate	Prior to elution of the filter paper samples, 200 μL of internal standard containing 0.150 mmol/L MMA-D3 and 0.075 mmol/L creatinine-D3 in water was deposited on each filter paper disk and left to dry for 2 h in ambient air. The 5 cm paper disk was, afterward, folded in half and deposited in a 20 mL glass vial. For the elution, 3 mL of NH_4_OH 0.01 mol/L was added to the glass vial, and then agitated for 10 min at 300 rpm on a model G2 gyratory shaker. For each filter paper eluate, 600 μL was filtered with a 1 mL, 0.2 μm water-wettable PTFE AcroPrep Advance 96-well filtration plate using an AcroPrep multi-well plate and vacuum manifold (Pall Corporation). The filtrate was retrieved in a 1 mL 96-well collection plate, which was covered with an XP-100 sealing film to prevent clogging of the needle seat by cellulose paper particles during the LC analyses.	LC: (1) Column:Charged surface hybrid C18 reversed-phase column; (2) Mobile phase: (A) MeOH + 0.1% FA and (B) H_2_O + 0.1% FA. MS: (1) Both positive and negative modes were used; (2) Monitor from *m/z* 117 to *m/z* 73 for MMA and from *m/z* 120 to *m/z* 76 for MMA-^13^C_4_; (3) MRM mode.	Intra- and inter-assay imprecisions were 4.8–5.2% and 4.9–9.0%, respectively. Recovery: 96.8–100.9%. 0.26 μmol/L. 0.86 μmol/L.	Analytes: MMA and creatinine. Validated sample types: dried urine in urine filter paper. Validated Applications: claimed suitable for vitamin B12 deficiency screening in older adults but no validations on relevant patients. Procedure step: claimed simple and user-friendly but requiring multiple steps and not fast, especially, a step of drying for 2 h; time-consuming. The method is claimed well suited for a future large-scale screening program of vitamin B12 deficiency in older adults, which is not validated on relevant patients and may be contrary to the results of Ma et al. [34], who reported urine MMA may not be suitable markers for assessing VB12 status. Not that cost-effective since requiring creatinine to be measured along with MMA, raising the cost.
Zheng et al. (2022, July) [38]	Xevo TQ-MSmicro mass spectrometer (Waters, Manchester, UK) equipped with a UniSprayTM interface and a Waters Acquity UPLC I-Class Plus system (Waters, Milford, MA)	Acetonitrile Methanol Formic acid Water	Clinical samples were centrifuged for 5 min at 3000× *g*. Samples (plasma/serum, 200 μL) were added to a plate consisting of 96 wells. The working solution of 500 μL acetonitrile consisting of 0.76 μmol/L MMA-D3 was added to the samples. The solutions were shaken for one minute, and then centrifuged for 10 min at 3000× *g*. The supernatant (300 μL) was added to another plate and evaporated with nitrogen gas for 30–45 min. The residual was reconstituted by 200 μL 0.2% formic acid in water and the plate was shaken for 1 min. Subsequently, the plate was centrifuged for 10 min at 3000× *g*, prior to LC-MS/MS analysis.	LC: (1) Column:Waters HSS-C18; (2) Mobile phase: (A) 0.2% formic acid in water and (B) 100% methanol. MS: (1) Negative modes were used; (2) Monitor from *m/z* 117 to *m/z* 73 for MMA and from *m/z* 120 to *m/z* 76 for MMA-^13^C_4_; (3) Multiple reaction monitoring(MRM) mode.	The within- and between-run CVs were 3–7%. Bias: from −1% to 8%. NA. 0.044 μmol/L.	Analytes: MMA only. Validated sample types: presented data indicates only on type was validated but not clear on which one was validated (claimed serum/plasma). Validated Applications: no validation on relevant patient for vitaminB12 deficiency assessment. Procedure steps: only consists of protein precipitation and centrifugations, but requires strict evaporation with nitrogen gas and reconstitutions; time-consuming. Required sample volume: 100 or 200 μL for plasma. Requiring at least 50 min for sample treatment. Their claimed requirements on steps: required to ensure that the acetonitrile has evaporated because the presence of this solution causes a double peak in the chromatogram. As mentioned above in Ma [34], the capability of acetonitrile in protein precipitation is weak, the direct injection of supernatant obtained by simple centrifugation may have potential risk of contaminating the mass spectrum or blocking the column; therefore, acetonitrile should not be selected for protein precipitation unless filter tubes is used for centrifugations. Due to more time required by evaporation with nitrogen gas, reconstitutions, and limited dryness devices, the method may not be convenient or suitable for large request volumes of MMA testing. Evaporation and reconstitution for several samples seems rapid but will be cumbersome for numerous samples.
Jin et al. (method in this study)	6500 Plus triple quadrupole mass spectrometer (AB Sciex, USA) coupled with an ExionLC™ AD ultra-high-performance liquid chromatography system (Applied Biosystems, CA, USA)	Methanol Acetonitrile Isopropanol Formic acid Water	50 μL of internal standard solution, 50 μL of samples, and 300 μL methanol were added to a 2.0 mL tube. The mixture were vortexed for 20 s and centrifuged at 148,000 rpm at 4 °C for 10–15 min. The upper phase of the mixture was poured into a new 2.0 mL tube and centrifuged at 148,000 rpm at 4 ℃ for 5 min. Fifty microliters of the upper phase were used for LC-MS/MS analysis. The injection volume was 1 μL.	LC: (1) Column:Shim-pack GIST-HP C18-AQ; (2) Mobile phase: (A) 0.1% formic acid and 5% isopropanol in water (100%A) and (B) no need. MS: (1) Negative modes were used; (2) Monitor from *m/z* 116.9 to *m/z* 72.8 for MMA and from *m/z* 120.9 to *m/z* 75.8 for MMA-^13^C_4_; (3) Multiple reaction monitoring(MRM) mode.	Intrarun, interrun, and total imprecisions were 1.42–2.69%, 3.09–5.27%, and 3.22–5.47%, respectively. The mean spiked recovery at three levels were 101.51%, 92.40%, and 105.95%, respectively. <0.058 μmol/L. 0.085 μmol/L.	Analytes: MMA only. Validated sample types: serum. Validated Applications: suitable for the MMA level evaluation in normal subjects and vitaminB12 deficiency patients. Procedure steps: only consists of protein precipitation and centrifugation, does not require evaporation, drying, or reconstitutions, etc. Required minor sample volume: 50 μL. Requiring less than 25 min for sample treatment. Less consumptions on materials and reagents, e.g., less tubes and less reagent volume used, etc. The mobile phase is salt-free, and it is easy to conduct equipment and column maintenance and startup. Organic solvents in only minor volumes are required for the mobile phase. (e.g., 1000 mL mobile phase only needs 50 mL isopropanol, 100 μL acetonitrile is used for rinse). More suitable for laboratories that receive a large number of requests for MMA testing.

**Table 2 diagnostics-12-02273-t002:** Spiked recovery and precision performance of the LC-MS/MS method for methylmalonic acid quantification.

Serum Pools	Mean Recovery ± SD				MMA Imprecision		
	Added, ng/g	Detected, ng/g	Recovery, %		Intra-Assay CV	Inter-Assay CV	Total CV
Level 1	0	29.62 ± 1.61	-		5.27	1.42	5.47
Level 2	14.95	44.69 ± 1.76	101.51 ± 5.74		3.09	2.69	4.10
Level 3	25.36	52.94 ± 2.30	92.40 ± 3.40		3.86	2.17	4.43
Level 4	37.20	68.92 ± 2.16	105.95 ± 1.95		3.83	1.55	3.22

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
