# Peer review of "A Rapid, Simple, Trace, Cost-Effective, and High-Throughput Stable Isotope-Dilution Liquid Chromatography–Tandem Mass Spectrometry Method for Serum Methylmalonic Acid Quantification and Its Clinical Applications"

_diagnostics, 2022, doi:10.3390/diagnostics12102273_

Round 1

Reviewer 1 Report (Previous Reviewer 1)

Good to go for publication. 

Author Response

Response to Reviewer 1 Comments

Point 1: Good to go for publication.

Response 1: Thank you very much for your thoughtful review and positive assessment.

Reviewer 2 Report (New Reviewer)

1. How this method better than other reported method. There arte number of literature reported the method dev and validation of this method. 

2. Degradation study must be performed.

3. Results must be more discussed with support of literature.

4. Conclude the important findings of the study.

5. Fonts in the Figure are very small.

Author Response

Response to Reviewer 2 Comments

Point 1: How this method better than other reported method. There arte number of literature reported the method dev and validation of this method.

Response 1: Thanks for your kindly reminding. We demonstrated the superiority over previously published methods in pages 5-6, and presented as “No complex and dangerous derivation reagents, costly/not always accessible ultrafiltration materials, nor processes that are time-consuming and laborious, such as evaporations, incubations, dryings, and reconstitutions are needed. The established method in this study demonstrated several advantages for MMA detection, e.g., convenient, environmentally friendly, economical, and more cost-effective for assessment of VB12 deficiency. As LC-MS/MS testing is mainly manual currently, such improvements are significant especially when numerous orders requested. Only several simple reagents in small volume were needed. Sample preparation can be completed in 20 minutes.”

Detailed information on the performance characteristics and method descriptions were shown in Table 1 A literature review of previous methylmalonic acid quantification based on the liquid chromatography-tandem mass spectrometry method (from 2000 to 2022).

Point 2: Degradation study must be performed.

Response 2: Thanks for pointing out this problem in the manuscript. MMA is a stable substance that has been reported in many studies (see reference [14,21-38] in the manuscript). Kushnir MM et al [1] reported that the stability of MMA exceeded three years in frozen samples and was unaffected by up to five freeze/thaw cycles. Fasching C et al [2] showed that MMA standard solution is stable for 1 year when stored at -10 to -20°C. Since many studies have indicated that MMA is a stable by-product of the vitamin B12 metabolic pathway, we did not perform the degradation study in our present manuscript. The stability study of MMA serum and the standard solution will be performed in the future.

References:

[1] Kushnir MM, Komaromy-Hiller G, Shushan B, Urry FM, Roberts WL. Analysis of dicarboxylic acids by tandem mass spectrometry. High-throughput quantitative measurement of methylmalonic acid in serum, plasma, and urine. Clin Chem. 2001 Nov;47(11):1993-2002. PMID: 11673368.

[2] Fasching C, Singh J. Quantitation of methylmalonic acid in plasma using liquid chromatography-tandem mass spectrometry. Methods Mol Biol. 2010;603:371-8. doi: 10.1007/978-1-60761-459-3_36. PMID: 20077089.

Point 3: Results must be more discussed with support of literature.

Response 3: We gratefully thank you for the helpful suggestion. The characteristics of analytical procedures in published methods and the established methods were discussed in the Introduction. We discussed the advantages of convenient operation, low cost, rapid results, and so on. The modifications were highlighted in yellow and presented as “Different from the previous path of improvement, we took advantage of a simple mobile phase strategy to improve simple MMA detection. This method requires fewer reagents and smaller sample. No complex and dangerous derivation reagents, costly/not always accessible ultrafiltration materials, nor processes that are time-consuming and laborious, such as evaporations, incubations, dryings, and reconstitutions are needed. The established method in this study demonstrated several advantages for MMA detection, e.g., convenient, environmentally friendly, economical, and more cost-effective for assessment of VB12 deficiency. As LC-MS/MS testing is mainly manual currently, such improvements are significant especially when numerous orders requested. Only several simple reagents in small volume were needed. Sample preparation can be completed in 20 minutes.”

Point 4: Conclude the important findings of the study.

Response 4: Thanks for the helpful suggestion. We have rewritten the Conclusions and emphasized the superiority and importance of the established method. The revision has been highlighted in yellow and presented as “This LC-MS/MS method can act as an easy assay and provide fast report results to evaluate the MMA status for vitamin B12 deficiency patients, especially applicable for large-scale MMA testing.”

Point 5. Fonts in the Figure are very small.

Response 5: Thanks for pointing out this issue in the manuscript. To make it easier for readers to see, we can provide clear original photos for the editors for better presentation.

This manuscript is a resubmission of an earlier submission. The following is a list of the peer review reports and author responses from that submission.

Round 1

Reviewer 1 Report

Jin et al. present a thorough study on quantifying serum methylmalonic acid, a measurable indicator of vitamin B12 deficiency with its upregulation in the blood. The study is particularly important because the complementary immunoassay for vitamin B12 suffers from cross-reactivity and often fails to exhibit the actual level. I do not have any objection to this study. Indeed, the mass spectrometry-based diagnostic is the future of laboratory medicine, which authors are directed to an important article: ACS Omega 2020, 5, 2041−2048. 

Author Response

Response to Reviewer 1 Comments

Point 1: Comments and Suggestions for Authors

Jin et al. present a thorough study on quantifying serum methylmalonic acid, a measurable indicator of vitamin B12 deficiency with its upregulation in the blood. The study is particularly important because the complementary immunoassay for vitamin B12 suffers from cross-reactivity and often fails to exhibit the actual level. I do not have any objection to this study. Indeed, the mass spectrometry-based diagnostic is the future of laboratory medicine, which authors are directed to an important article: ACS Omega 2020, 5, 2041−2048.

Response 1: Dear Referee, thank you very much. We’re extremely grateful for you spending time carefully reviewing our manuscript and providing nice comments for us. We have carefully considered your comments and enjoyed the important article you shared. Besides, we strongly agree with you that mass spectrometry-based diagnostic is the future of laboratory medicine. Through this technology, our lab has established reference methods or candidate reference methods for over 20 biochemical test items such as metabolites, blood lipids, enzymes, steroid hormones, inorganic ions, and glycosylated hemoglobin (https://www.nccl.org.cn/mainEn). These MS-based methods have played important and indispensable roles in our daily harmonization and standardization work, .., etc. It is not an exaggeration to say that the era of MS-based diagnostic in laboratory medicine perhaps has begun for a time.

Methylmalonic acid is such an example since it can’t be measured by routine economic immunoassays now but it can be accurately measured by the Lc-ms method. The article you shared exactly helps us to better illustrate the MMA detecting status, we cited an interesting concept from the shared article to describe it, i.e., “mass spectrometry-based disease biomarker”, MMA is such a biomarker! Thank you again for your nice comments and advice.

Thank you very much for reading our replies.

Best regards!

Lizi Jin

National Center for Clinical Laboratories,

Chinese Academy of Medical Sciences and Peking Union Medical College, Beijing.

Reviewer 2 Report

The manuscript focuses on a subject already explored in the field and does not present relevant novelties. There are several published papers on MMA measurements by LC-MS/MS, derivatized, non-derivatized, more and less complicated sample extraction, etc. The present study is not badly conducted, but does not bring significant originality, so my recommendation is to reject it.

It is said that:

·         “One of the reasons is that the majority of these methods are still complex and require dangerous reagents and consumables which were not easily available [10,15,21-29]. They are still time-consuming and laborious procedures (e.g., derivatization, and/or multi-step solid-phase extraction (SPE), and multidrying), making the test low-throughput and expensive”

·         “Our procedure not only eliminates derivatization and multistep SPE but is also free of drying (a common necessary time-consuming step before LC- MS/MS analysis)”

For example, the following one don’ include derivatization step and has a sample preparation procedure similar to that presented in the manuscript:

·         Lakso HA, Appelblad P, Schneede J.Clin Chem. 2008 Dec;54(12):2028-35. doi: 10.1373/clinchem.2007.101253. Epub 2008 Oct 9.PMID: 18845771

Even the procedure described by la Marca G, Malvagia S, Pasquini E, Innocenti M, Donati MA, Zammarchi E. (Clin Chem. 2007 Jul;53(7):1364-9. doi: 10.1373/clinchem.2007.087775. Epub 2007 May 17.PMID: 17510301) for DBS, has a sample preparation similar to the one here described and is widely used by several metabolic laboratories also for plasma samples.

Even the procedure described by la Marca G, Malvagia S, Pasquini E, Innocenti M, Donati MA, Zammarchi E. (Clin Chem. 2007 Jul;53(7):1364-9. doi: 10.1373/clinchem.2007.087775. Epub 2007 May 17.PMID: 17510301) for DBS, has a sample preparation similar to the one here described, and is widely used by several metabolic laboratories also for plasma samples.

Author Response

Response to Reviewer 2 Comments

Point 1: Comments and Suggestions for Authors

The manuscript focuses on a subject already explored in the field and does not present relevant novelties. There are several published papers on MMA measurements by LC-MS/MS, derivatized, non-derivatized, more and less complicated sample extraction, etc. The present study is not badly conducted, but does not bring significant originality, so my recommendation is to reject it.

Response 1: Dear Referee, thank you very much for spending time reviewing our manuscript, and thank you for your helpful comments. We have very carefully considered your comments and made relevant revisions and explanations based on your comments. The manuscript presentation has been improved in aspects of novelties and contributions. We think you will be satisfied with this revision.

To better illustrate the novelties and contributions of our manuscript, as mentioned above, we have summarized method details and their applications of all published articles (from 2000 to 2022) in Table 1. You may see from table 1:

MMA measurement is important and the development or improvement of the MMA detection method continues till the latest 5 years, even till 2021 (e.g., “High-throughput analysis of total homocysteine and methylmalonic acid with the efficiency to separate succinic acid in serum and urine via liquid chromatography-tandem mass spectrometry. J Chromatogr B Analyt Technol Biomed Life Sci. 2022;1193:123135”), it has a “similar” procedure to the two simple method examples you mentioned and some simple methods in Table 1 (they are mainly similar in the method principle, i.e., protein precipitation). There are many existing LC-MSMS methods for MMA and this recently published method represents a variation of them, actually, these simple methods are variations of each other. Despite all of this, our established methods have more clear advantages over this recently published simple method, i.e., simpler (not require drying step and residue reconstitution step) and more cost-effective. MMA can’t be measured by immunoassays while the total homocysteine (tHCY) can, so the measurement of tHCY by LC-MS seems to be unnecessary, expensive, and time-wasting (1. costs for the expensive isotope-labeled internal standard, pure standard materials, and DTT reagents used for amino acids analysis, and costs for method development and validation for tHCY; 2. time for integrating and/or checking chromatographic peaks to quantification for tHCY, etc), although the simultaneously measuring tHCY and MMA may avoid multiple blood collection. Besides, LC-MS/MS assays are manually operated, and any improvements can make difference. Getting rid of both the drying step and residue reconstitution step can be very important since MMA is proven to be a really significant diagnostic biomarker and the test order volume is increasing with its diagnostic value and simple MMA tests be known to more clinicians. In the case of large-volume test orders, clinical laboratories, clinicians, and importantly, patients will benefit from the savings of costs, time, and steps. And we did further useful improvements in these aspects. Each method has its own unique advantages, both theirs and ours.

You may also see there are laboratories measuring MMA combined with other biomarkers, like tHCY, Methionine, 2-Methylcitric Acid, and 3-OH-propionic acid. However, it is not that the more analytes they measure, the better the method. For newborn screening, maybe the more the better since these biomarkers of inborn errors/organic acidemia are mostly mass spectrometry-based disease biomarkers. While for other immunoassay-measurable indicators, it involves many considerations, e.g., the cost-benefit, which is an important concept in the era of MS-based diagnostic technologies and Diagnosis Related Groups (DRG). As we mentioned above, each method has its unique advantages or disadvantages, sometimes, the advantages are disadvantages, and the disadvantages are advantages. It is up to the procedures, clinical applications, and clinical settings, that is, when considering the novelties, and especially, the contributions, perhaps it should give attention to not only procedure simplicity but the clinical applications and clinical settings. MMA is a special biomarker who have diagnostic value in both vitaminB12 deficiency (as the manuscript written, “high prevalence across many populations”) and rare inborn errors (e.g., methylmalonic acidemia, our recently published META analysis showed the prevalence of it was 1.14 per 100,000 newborns). The simultaneous quantification of MMA and biomarkers like Methionine, 2-Methylcitric Acid, 3-OH-propionic acid are suitable for the inherited metabolic laboratories for inborn errors screening but the measurement of these biomarkers were unnecessary and irrelevant for the vitaminB12 deficiency diagnosis and not applicable for most routine clinical laboratories. Because in these method articles, they validated their methods in inborn error patients but they haven’t validated their methods’ clinical utility in vitaminB12 deficiency patients. Not only this, they’re low cost-effective for vitaminB12 deficiency diagnosis as mentioned before because the aim of such method is not for vitamin deficiency. The simultaneous quantification of MMA and biomarkers also suffer the loss of performance for MMA detection (you can see Table 1 for it).

From table 1, You may also see methods measuring MMA only. When compared with the aspects of sample volume, steps, hard-to-get consumable materials or reagents, and the consumption of reagents, solvent preparation, method performance, and clinical application validations, you may find we did make significant improvements, making it more suitable for labs which have numerous MMA testing orders.

Again, each method has its unique advantages or disadvantages, the advantages are disadvantages, and the disadvantages are advantages. Thus, we do not describe the advantages or disadvantages of each available methodologies concerning the measurement of MMA, instead, we use the word “characteristic” as the sum of advantages and disadvantages.

Based on the above statements and the following points, we deem that the subject is worth an updated exploration and it has unique novelties and contributions. Your comments are helpful, and according to your following comments, we have made revisions to better promote the introduction of the significance of our method (you can see it from the revised table and manuscript). We’re thankful for you reading to this point, it does take time, please continue to see the next points, and hope you’ll be satisfied with our revisions.

Point 2: It is said that: “One of the reasons is that the majority of these methods are still complex and require dangerous reagents and consumables which were not easily available [10,15,21-29]. They are still time-consuming and laborious procedures (e.g., derivatization, and/or multi-step solid-phase extraction (SPE), and multidrying), making the test low-throughput and expensive”, “Our procedure not only eliminates derivatization and multistep SPE but is also free of drying (a common necessary time-consuming step before LC- MS/MS analysis)” “For example, the following one don’ include derivatization step and has a sample preparation procedure similar to that presented in the manuscript: Lakso HA, Appelblad P, Schneede J.Clin Chem. 2008 Dec;54(12):2028-35”.”Even the procedure described by la Marca G, Malvagia S, Pasquini E, Innocenti M, Donati MA, Zammarchi E. (Clin Chem. 2007 Jul;53(7):1364-9.) for DBS, has a sample preparation similar to the one here described and is widely used by several metabolic laboratories also for plasma samples.”

Response 2: Dear Referee, thank you very much for this comment. We have carefully considered the two examples you take. We guess you would like to use these two examples to point out these sentences are not suitable to demonstrate the detection status of the MMA test since there were many published simple methods. We have re-written this paragraph (the Introduction) based on your comments and table 1.

In addition, the two examples mentioned in your comments do not need the derivatization step, but their method is not suitable for vitaminB12 deficiency and large-volume MMA testing. In la Marca [24], we found their method low-sensitivity, i.e., the highest LOQ (4.2 μmol/L) and LOD (1.9 μmol/L) compared with other methods and ours (<0.085 μmol/L), while reference intervals for MMA usually < 2 μmol/L and MMA levels in vitamin B12 patients are not as evidently high as the levels in inherited disorder patients (difference in this two levels can be tens to hundreds-folds, stated in the Discussion), indicating this simple method may not suitable for the investigation of populations with normal to deficient levels of vitamin B12. In Lakso HA et al [25] (or Table 1), their LC-MS/MS run time is higher than other methods (1-4 min), an injection requires 10 minutes while MMA is presented in 3 min and the remaining time is used to rinse to reduce the carryover effect. It’s kind of time-consuming when there are lots of orders. Also, their mobile phase used 100 mmol/L ammonium acetate buffer, which can easily cause contamination or scrapping of the LC or MS or column! While our mobile phase strategy is simple and salt-free, allows the minor carryover and rapid analysis of MMA, and has fewer requirements for the maintenance of equipment and column after use or during use.

Since quantification methods established based on LC-MS/MS are mostly manually performed in laboratories, tiny improvements in the reduction of steps, materials, solvent preparation, costs, and time can be significantly beneficial for technicians, clinicians, and importantly, patients. Based on the above, we deem that our method has its unique advantages and the manuscript has its unique contributions to MMA fast quantification.

Point 3: Open review: “Are all the cited references relevant to the research? Must be improved.” And “Moderate English changes required”.

Response 3: Dear Referee, thank you very much for this comment. We have carefully checked the contents of all references and our manuscript to ensure all references are relevant to the contents of the manuscript. Besides, the first manuscript you mentioned “Lakso HA, Appelblad P, Schneede J. Quantification of methylmalonic acid in human plasma with hydrophilic interaction liquid chromatography separation and mass spectrometric detection. Clin Chem. 2008;54(12):2028-35” had already been cited in our manuscript; the manuscript “la Marca G, et al. Rapid 2nd-tier test for measurement of 3-OH-propionic and methylmalonic acids on dried blood spots: reducing the false-positive rate for propionylcarnitine during expanded newborn screening by liquid chromatography-tandem mass spectrometry. Clin Chem. 2007;53(7):1364-9” was carefully read and added to the manuscript (see Table 1). Right as you said, the two examples have simple procedures for MMA measurements, nevertheless, the method in our manuscript has its unique advantages and it’s different from these methods, the demonstrations for this are illustrated in the following response and also the manuscript (please continue to see the next points). Now, at this point, we have confirmed all the cited references are relevant to the research and they’re necessary literature evidence supporting the manuscript. A new literature search for all published MMA methods with their essential information is summarized in Table 1 (from 2000 to 2022), too. (PubMed: LC-MS/MS AND MMA)

The revised manuscript (including tables and figures) has been checked by an English professional who has enriched experience in article writing and publishing (Dr. Xueyan Han, Department of Medical Statistics, Peking University First Hospital, Beijing, China).

Thank you very much for reading so long. Hope we give you satisfactory revisions.

Best regards!

Lizi Jin

National Center for Clinical Laboratories,

Chinese Academy of Medical Sciences and Peking Union Medical College, Beijing.
